# Effect of Platelet-Rich Plasma Augmentation on Endoscopy-Assisted Percutaneous Achilles Tendon Repair

**DOI:** 10.3390/jcm11185389

**Published:** 2022-09-14

**Authors:** Chun-Yu Hung, Shih-Jie Lin, Chia-Yi Yeh, Wen-Ling Yeh

**Affiliations:** 1Department of Orthopedic Surgery, Chang Gung Memorial Hospital, Yunlin 638, Taiwan; 2Department of Orthopedic Surgery, Jen-Ai Hospital, Taichung 412, Taiwan; 3Department of Orthopedic Surgery, New Taipei Municipal TuCheng Hospital, Chang Gung Memorial Hospital, New Taipei City 236, Taiwan; 4Department of Chemical Engineering, National Tsing Hua University, Hsinchu 300, Taiwan; 5Department of Athletic Training and Health, National Taiwan Sport University, Taoyuan 333, Taiwan; 6Department of Orthopedic Surgery, Chang Gung Memorial Hospital, Linkou 333, Taiwan; 7Department of Orthopedic Surgery, Lotung Poh-Ai Hospital, Yilan 265, Taiwan

**Keywords:** Achilles tendon rupture, platelet-rich plasma

## Abstract

Background: Achilles tendon ruptures are one of the most common sports injuries. Recently, platelet-rich plasma (PRP) has been widely used in tendon-related disorders to enhance tendon healing. However, studies regarding PRP treatment in Achilles tendon rupture show inconsistent results. The purpose of this study was to evaluate the effectiveness of PRP in patients with acute Achilles tendon rupture treated with endoscopy-assisted percutaneous repair. Methods: A total of 62 patients with acute Achilles tendon rupture treated with surgical repair from January 2014 to December 2018 were enrolled in this study. Surgical repair in conjunction with PRP augmentation after surgery was classified as the PRP group. Surgical repair without PRP augmentation was classified as the non-PRP group. All patients were followed up at least 2 years post-operation. The outcomes were evaluated on the basis of rate of return to sports, time to return to play, Achilles Tendon Total Rupture Score (ATRS), calf circumference ratio, ankle range of motion (ROM) and complications following surgery. Results: At 2-year follow-up, the ATRS score was not significantly different between groups (*p* = 0.8), but the ATRS score in both groups improved with time. Rate of return to sports and time to return to play were not different between the two groups (*p* = 1.00). Moreover, calf circumference ratio and ankle ROM were evaluated at 6-month, 12-month, 18-month and 24-month follow-ups. At 6 months, the PRP group had better ankle ROM (*p* = 0.003) and a higher calf circumference ratio (*p* = 0.011); however, at the 24-month evaluation, there were no between-group differences regarding calf circumference ratio, ankle dorsiflexion and plantarflexion (*p* > 0.05). Conclusion: We show that PRP augmentation in Achilles tendon surgery did not yield superior functional and clinical outcomes. Therefore, clinicians should inform patients of the above information when undergoing Achilles tendon surgery and offer correct expectations to family and patients regrading PRP treatment.

## 1. Introduction

The Achilles tendon is the strongest and largest tendon in the human body, subjected to the tensile loads up to ten times body weight during running [1]. However, the Achilles tendon is the most commonly injured tendon in the lower extremity and its incidence has increased in recent years, especially among individuals active in sports [2]. Achilles tendon ruptures necessitate year-long postoperative rehabilitation, leaving a 10% to 30% reduction in functional calf strength and endurance [3,4,5,6,7]. This injury leads to long-term sequelae and many patients are unable to return to preinjury levels of sports and physical activity. Treatment options include conservative treatment and surgical repair [7]. Surgery is preferred for young active individuals and those with high functional demands, thereby accelerating the healing process and reducing the re-rupture rate compared with non-operative treatment [8].

Recently, platelet-rich plasma (PRP) has gained increasing attention in regeneration medicine and been widely used in clinical treatment of injured tendons [9,10,11]. PRP contains a variety of growth factors, such as transforming growth factor β, platelet-derived growth factors, vascular endothelial growth factors and insulin-like growth factor I, that decrease inflammation and promote the healing process [12,13,14]. For this reason, PRP are now explored in an attempt to minimize the morbidity caused by Achilles tendon rupture. Whether PRP can promote the healing of ruptured Achilles tendons is a topic of great interest among sports medicine surgeons. Flourishing studies have been published on this field, but results are conflicting regarding the safety and efficacy of PRP [11,15,16,17,18,19].

Apart from its potential tissue regeneration properties, recent studies even reported that PRP possesses antimicrobial activities. Mariani et al. [20] showed that the bacterial growth of five strains (*E. coli, S. aureus, K. pneumoniae*, *P. aeruginosa* and *S. faecalis*) can be inhibited by the presence of PRP during in vitro culture. Serraino et al. [21] proved that the routine use of PRP could significantly reduce the incidence of deep sternal wound infection in cardiac surgery without any adverse reaction to it. However, there is no clear evidence about PRP’s antimicrobial activities in repaired Achilles tendons.

With this conflicting background, we performed a retrospective study to answer the following questions: Does PRP application result in (1) earlier return to sports; (2) a difference in infection rate compared with the non-PRP group; (3) a greater improvement in functional outcomes in patients with repaired Achilles tendons.

## 2. Patients and Methods

### 2.1. Study Design

We retrospectively studied prospectively collected data, analyzing the association between PRP augmentation and patient outcomes in Achilles tendon repair. This study protocol was approved by the Institutional Review Board of the Chang Gung Memorial Hospital (reference no.201802011A3). All surgical procedures took place at a single institution and a single experienced surgeon (W.L.Y.) participated in all of the operations.

Demographic data, including age, gender, body mass index (BMI), tobacco use, sports activity and time between injury to operation, were collected. We classified patients into a PRP augmentation group (32 patients) and a non-PRP augmentation group (30 patients). All patients received endoscopic-assisted Achilles tendon repair. In the PRP augmentation group, first, PRP was injected into the paratenon sheath under the guidance of endoscopy before wound closure. Second, PRP injection was given 2 weeks postoperatively with the same protocol under the guidance of ultrasonography.

### 2.2. Eligibility

From January 2014 to December 2018, 72 consecutive adult patients (59 males and 3 females) who sustained acute Achilles tendon ruptures treated with endoscopy-assisted percutaneous repair were recruited into this study. All patients had injuries related to sports, including basketball, badminton, soccer, jogging and other sports. We excluded patients who had chronic ruptures, re-ruptures, a steroid injection-related injury, a bony Achilles tendon avulsion, an acute trauma more than 10 days prior, who were under cortisone therapy, whose age was >60 years, and who had had a contralateral leg injury in the past. All patients were diagnosed with an Achilles tendons rupture via clinical histories, palpable tendon defects, a positive Thompson test [22] and the use of ultrasonography. Overall, 62 patients were eligible for inclusion in the study.

### 2.3. Surgical Technique

Under general or spinal anesthesia, patients were placed in the prone position with pneumatic tourniquet applied. All surgical procedures were performed by a single surgeon using the endoscopy-assisted percutaneous repair technique and modified Bunnell sutures [23].

### 2.4. Platelet-Rich Plasma

The autologous PRP was prepared using the Regen Kit (REGENLAB, RegenACR-C Classic, Le Mont-sur-Lausanne, Switzerland) and followed a standardized preparation procedure. Approximately 4 mL of PRP was used in each injection. First, injection of PRP was performed under the guidance of endoscopy during surgery. 2 weeks after surgery, a second PRP injection was performed in an outpatient clinic. Using a sterile technique, ultrasound-guided PRP was injected into the peritendineum of ruptured gap using a 23 g needle. A local anesthetic agent was not used for the injection procedure.

### 2.5. Rehabilitation

Postoperatively, a below-knee functional orthosis was used for 6 weeks at 45 degrees of plantar flexion. Patients were allowed to partially bear weight, as tolerated, using a single crutch in the first 6 weeks. Then, after 6 weeks, patients commenced a rehabilitation program, including range of motion exercises, and whirlpool and progressive resistance exercises. Active and passive ankle dorsiflexion were begun after 8 weeks. Rising on toes or heels was allowed 12 weeks post-operation, and limited sports activities were allowed after 3 months, with full loading recommended 6 months after surgery [23].

### 2.6. Outcome Assessment

Function outcomes were assessed during follow-up with the Achilles Tendon Total Rupture Score (ATRS). The ATRS is a patient-reported instrument with high reliability, validity and sensitivity for measuring outcome after treatment in patients with a total Achilles tendon rupture [24]. The ATRS comprised 10 items and a maximum score of 100 points reflected full function and no pain during activity; scores became progressively lower with worsening symptoms [24,25].

Moreover, ankle range of motion on the injured and uninjured sides, calf diameter of the injured and uninjured leg, status of tendon healing and Thompson test results were evaluated by two of the authors (S.J.L. and C.Y.Y). We also assessed patients’ ability to stand on tiptoes for 30 s, hop on one limb and perform repeated toe raises for 30 s [26].

Calf circumference was measured with patients sitting on the edge of the table with the knee flexed at 90 degrees and the lower legs relaxed [27]. Calf circumference was measured 10 cm below the tibia tuberosity and a reduction in calf circumference indicated muscle atrophy [16,27]. Meanwhile, the uninjured (contralateral) leg was measured to calculate side-to-side ratio.

### 2.7. Statistical Analysis

All analyses were performed using the SPSS statistical software, version 22.0 (SPSS Inc., Chicago, IL, USA). For categorial data, Fisher’s exact test was used for between-group comparisons. The Mann–Whitney U test was used to compare non-normally distributed numerical outcome data among the groups. Results were considered statistically significant at a *p* value < 0.05.

## 3. Results

### 3.1. Patient Characteristics

In total, 62 patients who met the inclusion criteria were enrolled and followed up for at least 2 years (range, 2.2–3.5 years). The demographic and preoperative details of all patients are shown in Table 1. There were no significant differences in age, body mass index and average interval between injury and surgery between the two groups. All patients could stand on tiptoe for more than 30 s, perform single-leg hopping, perform repeated toe raises for 30 s and have passive plantar flexion with the Thompson test at final follow-up.

### 3.2. Return to Work and Sports

Overall, 96.9% of patients in the PRP group and 96.7% of patients in the non-PRP group were able to return to work in 12 weeks after surgery. 100% of patients in both groups returned to work at 24-week follow-up. All but two patients (one patient in each group) returned to sports activity at 24-week follow-up (Table 2). These two patients developed wound infection that delayed the subsequent rehabilitation process.

### 3.3. ATRS Score

The ATRS score showed no difference between the two groups at 6, 12, 18 and 24-month follow-ups (Table 3). The mean ATRS score was 92.9 ± 3.2 for the PRP group and 92.7 ± 3.7 for the non-PRP group at 24-month follow-up. The ATRS score improved in both groups at each follow-up.

### 3.4. Calf Circumference and Ankle Range of Motion

Patients in the PRP group had a higher calf circumference ratio compared with the non-PRP group at 6-month follow-up. There were no between-group differences in side-to-side calf circumference ratio at 12-month, 18-month and 24-month follow-ups (Table 3). At 24-month follow-up, the calf circumference ratio was 95.8 ± 1.4 in the PRP group and 96.3 ± 1.7 in the non-PRP group. Calf circumference of the injured leg increased gradually with rehabilitation in both groups.

Patients in the PRP group had better ankle dorsiflexion at 6-month, 12-month and 18-month follow-ups compared with the non-PRP group, with statistically significant differences between the two groups. At 24-month follow-up, we found no significant difference in ankle dorsiflexion between the two groups (*p* = 0.071). For ankle plantarflexion, the PRP group had a better range of motion at 6-month follow-up (*p* = 0.003). However, there was no significant difference in ankle plantarflexion between the two groups at 12-month, 18-month and 24-month follow-ups (Table 3).

### 3.5. Complications

There were three complications in our series, with a complication rate of 4.8% (Table 4). Two wound infections developed, one in the PRP group and the other in the non-PRP group. These two patients recovered after antibiotic treatment and started rehabilitation thereafter. At 2-year follow-up, these two patients returned to their previous level of sports activity and reported an excellent ATRS outcome. One patient experienced sural nerve injury and improved gradually 3 weeks post-operation.

## 4. Discussion

The Achilles tendon, formed by the merging of the tendons of the gastrocnemius and soles, is the thickest and strongest tendon in the human body [8]. However, the Achilles tendon is the most commonly ruptured tendon, especially among male athletes [8,28]. Generally, in young, active individuals wishing to return to pre-injury activity, surgery remains the preferred treatment [8]. Studies have shown that surgical repair of ruptured Achilles tendons reduces the risk of re-ruptures compared with conservative treatment [8,29,30]. Nevertheless, Achilles tendon healing is a lengthy process due to poor blood supply and limited tissue turnover [31,32]. Basic researchers have proposed new concepts aiming to enhance the tissue regeneration potential in recent decades [33]. Among many therapeutic options to increase regeneration capacity, PRP has emerged as a promising and appealing adjunctive therapy for tendon healing [34]. From our previous laboratory work, platelet counts and several growth factors sustained or released a higher level than PRP obtained on the first day, during storage in a blood bank [35]. This implicated that PRP could continuously release growth factors from platelets or plasma at injection sites during regeneration. While the basic science shows promising results, the clinical benefits remain inconsistent. The goal of this study was to evaluate clinical outcomes of PRP augmentation in patients undergoing Achilles tendon repair. To our knowledge, this is the first study examining the efficacy of PRP augmentation in endoscopy-assisted Achilles tendons repair.

The literature regarding the effect of PRP on tendon healing is abundant. Kearney et al., showed that treatment with a single injection of intratendinous PRP did not reduce Achilles tendon dysfunction at 6 months among patients with chronic midportion Achilles tendinopathy [36]. Our data show that there was no beneficial effect of PRP augmentation after surgical intervention of acute Achilles tendon rupture. This is in line with a recent randomized, double-blind prospective study evaluating the effectiveness of PRP in non-surgically treated ruptured Achilles tendons in which no clinical benefits were seen in the ATRS score, heel-rise work, heel-rise height, tendon elongation, calf volume and ankle dorsiflexion ROM at 12-month follow-up [19]. Another randomized single-blind trial authored by Schepull et al. [16] reported that PRP augmentation may even have a detrimental effect on tendon healing, since no biomechanical benefit and a lower function score were noted in the PRP group compared with the non-PRP group. It is noteworthy that two studies reported positive effects of PRP on Achilles tendon rupture repair [15,18]. Critical appraisal of these contrasting results is necessary. Sanchez et al. [15] reported earlier return to training and better ankle ROM after augmentation of PRP in repaired Achilles tendons. Another study revealed that the PRP group had a significantly higher percentage of calf muscle strength than those in the control group at 3-month postoperative follow-up, implying that PRP augmentation may be an effective and feasible strategy to promote healing in repaired Achilles tendons [18]. Variation in growth factor concentrations of PRP and different treatment strategies may contribute to the inconsistent clinical efficacy. The encouraging results with PRP reported by Sanchez et al. [15] should be interpreted with caution because of small case numbers, with six participants in each group, and the PRP group was compared with a historical control group. Another prospective study by Zou et al. [18] suggests that PRP can serve as a biological augmentation to acute Achilles tendon rupture repair and therefore improves both short and mid-term functional outcomes. However, they used different surgical interventions for the rupture ends in the two groups, which may influence the final tendon length and functional outcomes. In the control group, the ruptured tendon was debrided before suturing, which may shorten the tendon, while in the PRP group, no tendon debridement was performed [18].

Resumption of sports and physical activity is a goal and an important component of successful treatment following Achilles tendon rupture [37]. However, there were large variations on return to play (RTP) rates among reported studies, ranging from 18.6% to 100% [38,39,40,41]. A recent systemic review and meta-analysis including 85 studies reported that 80% of patients were able to return to play and the average time to return to play was 6.0 months [37]. Whether PRP augmentations speed up return to sports following Achilles tendon ruptures remains the issue of much scrutiny. Filardo et al. [34] published a systemic review and failed to show earlier return to sports following PRP augmentation in ruptured Achilles tendons. In our results, RTP rates were 100% in both groups at 2-year follow-up. The average time to RTP in the PRP group and the non-PRP group were 6.5 and 6.8 months, respectively. Both RTP rates and time to RTP were not significantly different between the two groups.

Patient Reported Outcome Measures are self-reported questionnaires that have become cornerstone in evaluating the effectiveness of a treatment and patient satisfaction [42,43]. In the present study, we used the Achilles Tendon Rupture Score (ATRS), which is a patient-reported instrument with high reliability, validity and sensitivity for measuring the outcomes related to symptoms and physical activity after treatment in patients with a total Achilles tendon rupture [24]. However, it is difficult to compare functional outcomes with previous studies due to some studies using non-validated scores for patients undergoing Achilles tendon surgery. In our results, the ATRS in both groups improved with time and there was no significant difference between groups at 2-year follow-up (PRP group: 92.9 points; non-PRP group: 92.7 points). Calf circumference ratio and ankle ROM were comparable between the PRP group and the non-PRP group, which is in agreement with the results of previous studies [11,16,19]. Reduced calf circumference and ankle ROM were shown in both groups at 2-year follow-up, indicating that Achilles tendon rupture was a devastating injury, and that some degree of functional deficit was inevitable.

The clinical application of PRP has been widely studied for its potential effect on tendon injuries; however, the antibacterial property of PRP has been overlooked [34]. Increasing evidence suggests that PRP possesses antibacterial properties and it could prevent tissue infection [44]. Activated PRP releases several antimicrobial proteins, including connective tissue-activating peptide 3, platelet factor 4, thymosin β-4, platelet basic protein; fibrinopeptide A and fibrinopeptide B [45]. Several in vitro studies demonstrated that PRP can inhibit the adherence and growth of bacteria colonization [46,47,48,49]. Ahmed et al. [50], in a study comparing the healing rate of chronic diabetic foot ulcers among antiseptic ointment dressing-treated (control group) and autologous platelet gel-treated patients, demonstrated that PRP-treated group had a higher healing rate and a lower infection rate compared with the control group. A large-scale clinical study published by Patel et al. [51] reviewed 2000 patients during a 7-year period. They found that the use of PRP can prevent sternal wound infections and reduce readmission rates and costs following cardiac surgery, and they recommended routine use of PRP for all patients undergoing sternotomy for cardiac surgical procedures. In our study, wound infection rates in the PRP group and the non-PRP group were 3.1% and 3.3%, respectively (Table 4). This result fails to show any antibacterial effect of PRP on repaired Achilles tendons. As there were relatively small case numbers in our study, higher patient numbers are necessary to achieve adequate statistical power to verify this conclusion.

Complications following repair of ruptured Achilles tendons may delay tendon healing and function recovery. Stavenuiter et al. [52] retrospectively analyzed 615 adult patients receiving operative repair for an acute Achilles tendon rupture and reported an overall 11.7% complication rate, with the three most adverse events including wound problems (5.2%), venous thromboembolic events (3.6%) and sural nerve injury (2.0%). Moreover, they also identified advanced age (odds ratio, 1.04, *p* = 0.007) and active tobacco use (odds ratio, 3.20, *p* = 0.007) as being independently predictive for postoperative complications. Our results contrast with Stavenuiter et al. [52], who found minimally invasive repair (13.2%) had a higher complication rate than open repair (11.6%). Another systemic review found that minimally invasive surgery results in a lower rate of complications than open surgery, with a comparable functional outcome to traditional open procedures [53]. In our series, the incidence of wound problems and sural nerve injuries were 3.2% and 1.6%, respectively.

Nicotine exposure has detrimental effects of human health. From an orthopedic surgeon’s perspective, nicotine has been shown to impair tendon/ligament healing, delay fracture union, increase infection rate and worsen arthroplasty outcomes [54,55,56]. In our series, 2 of the 62 patients delayed return to pre-injury activity level due to wound infection (Table 3 and Table 4). Notably, these two male patients were heavy smokers and demonstrated non-adherence to the postoperative rehabilitation process. Chema et al. [56], in an animal study exploring the effect of nicotine on intra-substance tendon healing, reported that rats exposed to nicotine demonstrated decreased vascularity, greater alternation in gait mechanics and increased passive ROM of the ankle joint. Biomechanically, the nicotine tendons failed at lower maximum loads, were less stiff, had smaller cross-sectional areas and had altered viscoelastic properties [56]. Bruggeman et al. [57] reviewed complications of Achilles tendon repair and reported significant risk factors for development of wound complications included tobacco use (*p* < 0.0001), steroid use (*p* = 0.0005) and female sex (*p* = 0.04). As a result, at our institution, patients were counseled on smoking cessation prior to undergoing repair of ruptured Achilles tendons.

Some limitations should be addressed in this study. First, case numbers in this study are relatively small and higher patient numbers are necessary to achieve adequate statistical power to draw a firm conclusion. Second, we did not measure mechanical properties of healed Achilles tendons, such as modulus of elasticity and tendon strain under loading. Further studies are necessary to elucidate the effect of PRP on the biomechanical properties of repaired Achilles tendons. Third, selection of the PRP group was not randomized and was judged by the treating surgeon.

## 5. Conclusions

We found that PRP augmentation for the treatment of repaired Achilles tendons may not be beneficial. At 2-year follow-up, the ATRS score and ankle ROM are comparable between the PRP and non-PRP groups. Moreover, PRP augmentation does not reduce infection rate, increase return to play rate and facilitate quicker return to sports. Based on our data, the application of PRP in repaired Achilles tendons should not be used indiscriminately unless further evidence proves its efficacy. Clinicians should inform patients of the above information when undergoing Achilles tendon surgery and offer correct expectations to family and patients regrading PRP treatment.

## Figures and Tables

**Table 1 jcm-11-05389-t001:** Demographic data of patients with acute Achilles tendon rupture.

	PRP	Non-PRP	*p* Value
Number of patients	32	30	
Mean age (SD, range), y	37.5 (8.7, 22–50 years)	39.6 (8.5, 20–53 years)	0.95
Mean BMI (SD), kg/m^2^	24.8 (1.26)	25.6 (1.03)	0.059
Sex, *n* (%)			
Male	30 (93.8)	29 (96.7)	1.000
Female	2 (6.2)	1 (3.3)	
Tobacco use, *n* (%)	1 (3.1)	1 (3.3)	1.000
Average interval between injury and surgery (range), day	6.3 (2–10)	5.9 (2–10)	0.25
Sport, *n* (%)			
Soccer	7 (21.9)	6 (20)	
Basketball	5 (15.6)	8 (26.7)	
Badminton	9 (28.1)	8 (26.7)	
Jogging	2 (6.3)	3 (10)	
Other sports	9 (28.1)	5 (16.6)	

BMI, body mass index; PRP, platelet-rich plasma; SD, standard deviation. Statistically significant difference (*p* < 0.05).

**Table 2 jcm-11-05389-t002:** Return to work and sports based on PRP.

	PRP (*n* = 32)	Non-PRP (*n* = 30)	*p* Value
Return to regular work, *n* (%)			
12 weeks ^a^	31 (96.9)	29 (96.7)	1.000
24 weeks	32 (100)	30 (100)	
1 year	32 (100)	30 (100)	
Return to sports, *n* (%)			
24 weeks ^a^	31 (96.9)	29 (96.7)	1.000
1 year	32 (100)	30 (100)	

^a^ Two patients have superficial wound infections needing further antibiotic treatment.

**Table 3 jcm-11-05389-t003:** Functional tests and scores for patients with repaired Achilles tendon.

	PRP (*n* = 32)	Non-PRP (*n* = 30)	*p* Value
ATRS			
6 months	78.6 ± 6.8	76.7 ± 6.9	0.271
12 months	83.4 ± 5.5	85.7 ± 4.9	0.106
18 months	85.9 ± 4.6	87.7 ± 4.3	0.164
24 months	92.9 ± 3.2	92.7 ± 3.7	0.804
Calf circumference ^a^, %			
6 months	86.9 ± 2.3	85.3 ± 2.3	0.011 *
12 months	91.9 ± 1.8	91.3 ± 2.2	0.336
18 months	93.0 ± 1.5	92.4 ± 2.1	0.294
24 months	95.8 ± 1.4	96.3 ± 1.7	0.131
Dorsiflexion ^b^, degree			
6 months	3.5 ± 0.6	4.1 ± 0.6	0.003 *
12 months	2.7 ± 0.6	3.1 ± 0.6	0.024 *
18 months	2.1 ± 0.6	2.5 ± 0.5	0.01 *
24 months	1.8 ± 0.5	1.5 ± 0.5	0.071
Plantarflexion ^b^, degree			
6 months	3.6 ± 0.7	4.2 ± 0.4	0.003 *
12 months	2.1 ± 0.7	2.5 ± 0.4	0.062
18 months	1.6 ± 0.6	1.8 ± 0.3	0.313
24 months	1.2 ± 0.5	1.4 ± 0.3	0.058

ATRS, Achilles Tendon Total Rupture Score. ^a^ Calf circumference is expressed as the ratio between the injured limb and the uninjured limb. ^b^ Ankle range of motion is presented as the difference from the uninjured limb. * Statistically significant difference (*p* < 0.05).

**Table 4 jcm-11-05389-t004:** Characteristics of patients developing complications.

Gender	Age, Years	Cause of Injury	Smoking	PRP Augmentation	Complications	ATRS ^a^
M	40	Soccer	yes	yes	superficial wound infection	95
M	42	Badminton	yes	no	superficial wound infection	93
M	44	Running	no	no	sural nerve injury ^b^	85

^a^ ATRS at 2-year follow-up. ^b^ Lateral foot numbness lasted for 3 weeks and improved afterward.

## Data Availability

The dataset supporting the conclusions of this article is available from the corresponding author on reasonable request.

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
