# Peer review of "Effect of Platelet-Rich Plasma Augmentation on Endoscopy-Assisted Percutaneous Achilles Tendon Repair"

_jcm, 2022, doi:10.3390/jcm11185389_

Round 1

Reviewer 1 Report

The authors present an interesting study which compares the usage of PRP vs. non-PRP for Achilles tendon healing in an operative setting with minimally-invasive surgery. In general, the topic is very relevant and the manuscript fits into the Journal of Clinical Medicine. However, in my opinion, it needs a revision. To me, the study design appears to be prospective, but the question of time point of study set-up and inclusion needs to be taken into account (see below).

Some questions and suggestions in detail:

2 I suggest to use a stronger and more factual title, including the time point and study results

49: PRP contained or contains? Confusing statement.

51: Do all of these factors decrease inflammation and promote a healing process? Which healing processes? I suggest to give the reader a more detailed pathophysiological context. 

87 Were there some patients with a contralateral injury in the past included in the study?

93 I assume for blood arrest? I suggest to specify.

99 Was the PRP placed underneath the peritendineum in the area of rupture or just around the tendon? And were measured taken to secure the PRP at the area of injection? How big was the volume? I suggest to specify it for higher accuracy and repeatability.

106 Does “to mobilize to partial-weight bearing movement as tolerated” mean “allowance of partial weight bearing” or actual mobilization of the ankle joint (joint mobilization)? Please specify.

113-115 At which time-point were the parameters determined? Please state here. In the results we find different time points of 12 weeks, 24 weeks, 1 years, 6,12,18 and 24 months, and then a range of 2.2-3.5 years. This is very confusing, inconsistent, and not understandable, which data was collected when. Please describe the time points in the Methods section.

Additionally: If the data was collected with the purpose of this study and the patients consent for the study was collected before surgery in time, the study model would be prospective, not retrospective. 70-71 suggests otherwise, but the extent of data collection and amount of parameters suggest the intention of a prospective study before the time of surgeries.

116 Was the ATRS used in English or native language? Is there a validated native ATRS?

120 I suggest to use “uninjured” instead of “normal” throughout the manuscript.

121 How was “status of tendon healing” determined? There are no results shown regarding this parameter.

123 Were the patients instructed to hop on one limb after already 12 weeks?

148 Is that data based on the follow-up after 2.2-3.5 years?

166, 211, 215-218, 33-35 and others: check Syntax

287-289 Consider citing a meta-analysis instead of a single study regarding the comparison of open vs. minimally invasive techniques (e.g. Del Buono A et al. Minimally invasive versus open surgery for acute Achilles tendon rupture: a systematic review. Br Med Bull. 2014;109: 45–54).

308-311 Consider naming some current techniques like dynamometers (e.g. Bressel et al. Biomechanical Behavior of the Plantar Flexor Muscle-Tendon Unit after an Achilles Tendon Rupture. Am. J. Sports Med. 2001, 29, 321–326.) and un-invasive Shear Wave Elastography (e.g. Frankewycz, B. et al. Changes of Material Elastic Properties during Healing of Ruptured Achilles Tendons Measured with Shear Wave Elastography: A Pilot Study. International Journal of Molecular Sciences 21, 3427). 

15: “one of the most devastating sports injuries” sounds very drastic and doesn’t seem to be correct compared to joint-amputation, or para/tetraplegic injuries. Consider stating it differently.

Reviewer 2 Report

Comments to the Authors

The manuscript titled “Does Application of Platelet-Rich Plasma Improve Clinical Outcomes after Endoscopy-Assisted Percutaneous Repair of Achilles Tendon Rupture?” is a retrospective clinical trial which investigated any therapeutic effect of PRP augmentation during endoscopy-assisted Achilles tendon repair surgery.

The work is very interesting and shows aspects of novelty. Manuscript is well written and English language is appropriate. The experimental plan is properly designed and well developed; methods are clearly described and sufficiently detailed. The conclusions are well supported by the results.

Below are some minor issues that still need to be addressed:

·         The title should be rephrased, so that it is not presented as a question

·         In the Abstract, p values should be reported for each comparison between experimental groups, bot for significant and not significant differences

·         In the Patients and Methods Section there are some points that need careful revision:

- no information about the collection of informed consent from patients was reported. Authors are required to provide this information

- more details should be provided about the ATRS questionnaire administered to patients (e.g., which are the mentioned 10 items? Which was the score assigned for single item?)

- since the variability of the preparation protocols to obtain platelet-rich products as PRP is the main cause of their controversial clinical outcomes, it would be useful to describe how the PRP was prepared in this study. Moreover, it is not clear whether the PRP was autologous or was taken from a blood bank: please specify.

·         In the Discussion Section, some explanation should be speculated about the non-beneficial effect of PRP despite its high content in regenerative/anti-inflammatory/antimicrobial factors. Could a more structured product like PRF add some benefits for this clinical application? PRF is currently considered as the II generation of platelet-rich hemocomponents, representing an advancement of PRP therapeutic potential due to its 3D fibrin network which assures for better growth factor/cytokine entrapment and prolonged release. Do the Authors think that PRP could have been a not ideal choice of treatment in comparison with PRF for this kind of application? Please, discuss.
